



# Advanced climate model evaluation with ESMValTool v2.11.0 using parallel, out-of-core, and distributed computing

Manuel Schlund[1], Bouwe Andela[2], Jörg Benke[3], Ruth Comer[4], Birgit Hassler[1], Emma Hogan[4], Peter Kalverla[2], Axel Lauer[1], Bill Little[4], Saskia Loosveldt Tomas[5], Francesco Nattino[2], Patrick Peglar[4], Valeriu Predoi[6], Stef Smeets[2], Stephen Worsley[4], Martin Yeo[4], and Klaus Zimmermann[7]

[1]Deutsches Zentrum für Luft- und Raumfahrt (DLR), Institut für Physik der Atmosphäre, Oberpfaffenhofen, Germany
[2]Netherlands eScience Center (NLeSC), Amsterdam, the Netherlands
[3]Simulation and Data Lab Terrestrial Systems, Jülich Supercomputing Centre, Forschungszentrum Jülich, Jülich, Germany
[4]Met Office, Exeter, UK
[5]Barcelona Supercomputing Center (BSC), 08034, Barcelona, Spain
[6]NCAS-CMS, University of Reading, Reading, UK
[7]Swedish Meteorological and Hydrological Institute (SMHI), Folkborgsvägen 17, 601 76 Norrköping, Sweden

**Correspondence:** Manuel Schlund (manuel.schlund@dlr.de)

**Abstract.** Earth System Models (ESMs) allow numerical simulations of the Earth's climate system. Driven by the need to better understand climate change and its impacts, these models have become increasingly sophisticated over time, generating vast amounts of data. To effectively evaluate the complex state-of-the-art ESMs and ensure their reliability, new tools for comprehensive analysis are essential. The open-source community-driven Earth System Model Evaluation Tool (ESMValTool)

addresses this critical need by providing a software package for scientists to assess the performance of ESMs using common diagnostics and metrics. In this paper, we describe recent significant improvements of ESMValTool's computational efficiency, which allow a more effective evaluation of these complex ESMs and also high-resolution models. These optimizations include parallel computing (execute multiple computation tasks simultaneously), out-of-core computing (process data larger than available memory), and distributed computing (spread computation tasks across multiple interconnected nodes or machines). When

comparing the latest ESMValTool version with a previous not yet optimized version, we find significant performance improvements for many relevant applications running on a single node of a high performance computing (HPC) system, ranging from 2.6 times faster runs in a multi-model setup up to 25 times faster runs for processing a single high-resolution model. By utilizing distributed computing on two nodes of an HPC system, these speedup factors can be further improved to 3.2 and 36, respectively. Moreover, especially on small hardware, evaluation runs with the latest version of ESMValTool also require significantly

less computational resources than before, which in turn reduces power consumption and thus the overall carbon footprint of ESMValTool runs. For example, the previously mentioned use cases use 16 (multi-model evaluation) and 40 (high-resolution model evaluation) times less resources compared to the reference version. Finally, analyses which could previously only be performed on machines with large amounts of memory can now be conducted on much smaller hardware through the use of out-of-core computation. For instance, the high-resolution single-model evaluation use case can now be run on a machine with

only 16 GB of memory despite a total input data size of 35 GB, which was not possible with earlier versions of ESMValTool. This enables running much more complex evaluation tasks on a standard laptop than before.





# 1   Introduction

Earth system models (ESMs) are crucial for understanding the present-day climate system and for projecting future climate change under different emission pathways. In contrast to early atmosphere-only climate models, modern-day ESMs participating in the latest phase of the Coupled Model Intercomparison Project (CMIP6; Eyring et al., 2016) allow numerical simulations of the complex interactions of the atmosphere, ocean, land surface, cryosphere, and biosphere. Such simulations are essential in assessing details and implications of climate change in the future and are the base for developing effective mitigation and adaptation strategies. For this, thorough evaluation and assessment of the performance of these ESMs with innovative and comprehensive tools are a prerequisite to ensure reliability and fitness for purpose of their simulations (Eyring et al., 2019).

To facilitate this process, the Earth System Model Evaluation Tool (ESMValTool; Righi et al., 2020; Eyring et al., 2020; Lauer et al., 2020; Weigel et al., 2021; Schlund et al., 2023) has been developed as an open-source, community-driven software. ESMValTool allows for the comparison of model outputs against both observational data and previous model versions, enabling a comprehensive assessment of a model's performance. Through its core functionalities (*ESMValCore*; see Righi et al., 2020), which are completely written in Python, ESMValTool provides efficient and user-friendly data processing for commonly-used tasks such as horizontal and vertical regridding, masking of missing values, extraction of regions / vertical levels, calculation of statistics across data dimensions and/or data sets, etc. A key feature of ESMValTool is its commitment to transparency and reproducibility of the results. The tool adheres to the FAIR Principles for research software (Findable, Accessible, Interoperable and Reusable; see Barker et al., 2022), provides well-documented source code, detailed descriptions of the metrics and algorithms used, and comprehensive documentation of the scientific background of the diagnostics. Users can define their own evaluation workflows with so-called *recipes*, which are YAML files (https://yaml.org/, last access: 10 December 2024) specifying all input data, processing steps and scientific diagnostics to be applied. This provides a flexible and customizable approach to model assessment. All output generated by ESMValTool includes a provenance record, meticulously documenting the input data, processing steps, diagnostics applied, and software versions used. This approach ensures that the results are traceable and reproducible. By now, ESMValTool is a well-established and widely used tool in the climate science community used, for instance, to create figures of the Sixth Assessment Report (AR6) of the Intergovernmental Panel on Climate Change (IPCC; e.g., Eyring et al., 2021) and has been selected as one of the evaluation tools listed by the CMIP7 climate model benchmarking task team; see https://wcrp-cmip.org/tools/model-benchmarking-and-evaluation-tools/ and https://wcrp-cmip.org/cmip7/rapid-evaluation-framework/, last access: 10 December 2024).

Since the first major new release of version 2.0.0 in 2020, a particular development focus of ESMValTool has been to improve the computational efficiency of the tool, in particular in the ESMValCore package (Righi et al., 2020), which takes care of the computationally intensive processing tasks. This is crucial for various reasons. First, the continued increase in resolution and complexity of the CMIP models over many generations has led to higher and higher data volumes with the published CMIP6 output reaching approximately 20 PB (Petrie et al., 2021). Future CMIP generations are expected to provide even higher amounts of data. Thus, fast and memory-efficient evaluation tools are essential for an effective and timely assessment of current and future CMIP ensembles. Second, using existing computational resources more effectively reduces energy demand





and the carbon footprint of HPC and data centers, which are expected to have a steadily increasing contribution to the total global energy demand in the upcoming years (Jones, 2018) and thus increasingly contribute themselves to anthropogenic climate change. Finally, faster and more memory-efficient model evaluation reduces the need to use HPC systems for model evaluation, and allows using smaller local machines instead. This is especially relevant to the Global South, which still suffers
from limited access to HPC resources but at the same time is highly affected by climate change (Ngcamu, 2023).

In this study, we describe the optimized computational efficiency of ESMValCore and ESMValTool available in the latest release v2.11.0 from July 2024. Note that these improvements were not implemented within one release cycle, but rather developed in a continuous effort over the last years. The three main concepts we have used here are (1) parallel computing (i.e., performing multiple computation tasks simultaneously rather than sequentially), (2) out-of-core computing (i.e., processing
data that are too large to fit into available memory), and (3) distributed computing (i.e., spreading computational tasks across multiple interconnected nodes or machines). We achieve this by consequently making use of state-of-the-art computational Python libraries such as Iris (Iris contributors, 2024) and Dask (Matthew Rocklin, 2015).

This paper is structured as follows: Section 2 provides a technical description of the improvements in computational efficiency in the ESMValCore and ESMValTool packages. Section 3 presents example use cases to showcase the performance gain
through these improvements which are then discussed in Section 4. The paper closes with a summary and outlook in Section 5.

## 2   Improving computational efficiency

The main strategy for improving the computational efficiency in our evaluation workflow is by consistently using Dask (Matthew Rocklin, 2015), a powerful Python package designed to speed up computationally intensive tasks for array computations. An overview of Dask can be found in its documentation, for example in the form of a clear and concise 10-minutes
introduction (https://docs.dask.org/en/stable/10-minutes-to-dask.html, last access: 10 December 2024), which we summarize in the following.

Dask breaks down computations into individual *tasks*. Each task consists of an operation (e.g., compute maximum) and corresponding arguments to that operation (e.g., input data or results from other tasks). A key concept here is the partitioning of the involved data into smaller *chunks* that fit comfortably into the available memory, which enables Dask to also process data
that are too large to fit entirely into memory (*out-of-core* computing). A Dask computation is represented as a *task graph*, which is a directed acyclic graph (DAG) where each node corresponds to a task and each edge to the dependencies between these tasks. This task-based approach allows Dask to effectively parallelize computations (*parallel computing*) and even distribute tasks across multiple interconnected machines like different nodes of an HPC system (*distributed computing*). The task graph is then executed by the so-called *scheduler*, which orchestrates the tasks, distributes them efficiently across available *workers*
(components that are responsible for running individual tasks) and monitors their progress.

In practice, this may look as follows: first, the input data are loaded lazily (i.e., only metadata like the arrays' shape, chunk size, and data type are loaded into memory; the actual data remain on disk). Then, the task graph is built by using Dask structures and functions, for example, `dask.array` objects instead of `numpy` (Harris et al., 2020) objects. Again, this will



not load any data into memory, but merely sets up the tasks. Typically, the task graph consists of operations which only operate
on single chunks, and operations which aggregate the results from these "chunked" operations. For example, to calculate the
maximum of an array, an approach is to first calculate the maxima over the individual chunks, and then calculate the maximum
across the individual maxima. Finally, these tasks will be executed by the workers as directed by the scheduler, which will
eventually load the actual data into memory in small chunks and perform all desired computations defined in the task graph.

Dask offers different scheduler types, and choosing the most appropriate one for a specific application is crucial for optimum
performance. By default, Dask uses a simple and lightweight single-machine scheduler, which is, for array-based workflows
like our use cases here, based on threads. One major drawback of this scheduler is that it can only be used on a single machine
and does not scale well. A powerful alternative is the *Dask distributed scheduler*, which, despite having the word "distributed"
in its name, can also be used on a single machine. This scheduler is more sophisticated, allows distributed computing, and
offers more features like an asynchronous application programming interface (API) and a diagnostic dashboard, but adds
slightly more overhead to the tasks than the default scheduler and also requires more knowledge to configure correctly.

Whenever possible, we do not use Dask directly, but rely on the Iris package (Iris contributors, 2024) for data handling
instead. Iris is a Python package for analyzing and visualizing Earth science data. In the context of ESMValTool, a key
feature of Iris is the built-in support for data complying to the widely used Climate and Forecast (CF) conventions (https:
//cfconventions.org/, last access: 10 December 2024), which all input data of ESMValTool are expected to follow (see Schlund
et al. (2023) for details on this). Iris provides a high-level interface to Dask, for example, `iris.load()` will load data lazily,
or `iris.save()` will execute the Dask task graph and save the resulting output to disk.

Figure 1 provides a schematic overview of the usage of Dask in ESMValTool v2.11.0. To determine the relative runtime
contributions of the individual components, we used the py-spy sampling profiler (https://github.com/benfred/py-spy, last
access: 10 December 2024). The computationally most intensive part is the preprocessor, which is part of ESMValCore (dark
gray box in the figure), a dependency of ESMValTool that contains all of the tool's core functionalities including all data
preprocessing functions. Thus, enhancing the computational efficiency of ESMValCore's preprocessor through the use of Dask
yields the most significant overall performance improvements. In practice, this is achieved as follows: the data are lazily
loaded via Iris. After that, the individual preprocessor steps are executed (mandatory ones like data validation as well as the
custom ones like regridding, masking, etc.), which will build up the Dask task graph. This is done by using Dask or available
corresponding Iris functionalities wherever possible, but oftentimes also requires custom code. For example, when regridding
data to a higher resolution, the regridding preprocessor function needs to resize the input chunks before regridding so the output
chunk sizes fit into memory. Similarly, the multi-model statistics preprocessor function requires that all data share the same
calendar. Finally, a call to `iris.save()` loads the data into memory (in small chunks), executes the task graph and saves
the preprocessor output to disk. Implementing this pipeline in a computationally efficient way has been a massive effort over
the last years and is still ongoing. For example, as of 10 December 2024, 95 pull requests mention the key words "dask" or
"performance" in the ESMValCore repository. The Dask workflow within ESMValTool is highly customizable via configuration
files (see Appendices A and B), which allows selecting the type of scheduler, the number of workers, the amount of memory
used, etc. The Dask configuration needs to be adapted to the actual use case, i.e., the corresponding evaluation task and the



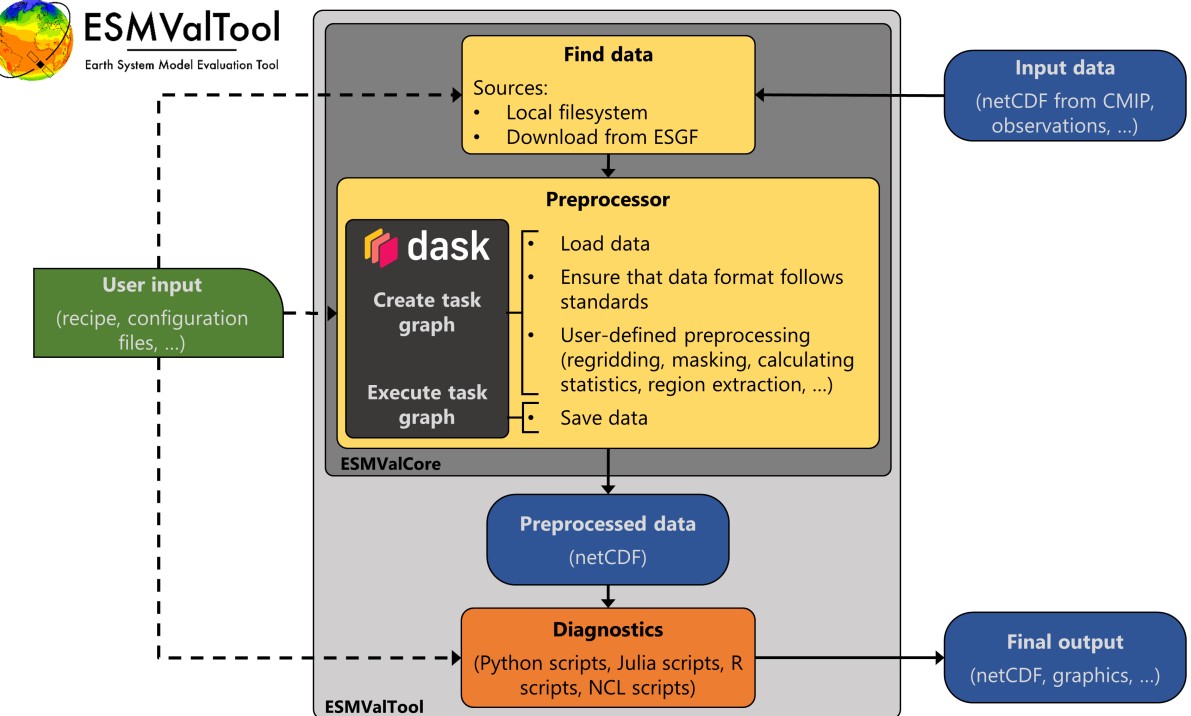

**Figure 1.** Schematic representation of ESMValTool (light gray box). Input data are located and processed by the ESMValCore package (dark gray box), which is a dependency of ESMValTool. The preprocessed data written by ESMValCore are then used as input by the scientific diagnostic scripts, which typically produce the final output of ESMValTool. User input is given via YAML files like the recipe and configuration files (green box). Since ESMValCore's preprocessor performs the majority of the computationally intensive operations, enhancing its computational efficiency with Dask yields the highest overall performance improvements for ESMValTool.

available hardware (memory, CPUs, etc.). Dask can also be used to improve the efficiency of scientific Python diagnostics in
ESMValTool (see orange box in Figure 1); however, since the bulk of the computationally intensive work is usually performed by the preprocessor, this is only improving overall performance significantly in a few selected cases where heavy computations are also done within a diagnostic. One major improvement in recent versions of ESMValTool, ESMValCore, and Iris is the added support for Dask distributed schedulers. As stated above, this scheduler type enables distributed computing, but can also provide substantial performance boosts on single-machine setups. Before introducing these changes, only the default
thread-based scheduler could be used.

## 3   Use cases

To showcase the improved computational efficiency of ESMValTool v2.11.0, this section provides examples of real-world use cases. The analyses we perform here measure the performance of ESMValTool along two dimensions: (1) different versions





**Table 1.** Overview of the different setups used to showcase the improved computational efficiency of ESMValTool v2.11.0. For the distributed scheduler, each worker uses 4 threads.

| Setup name | ESMValTool version | Number of nodes | Total memory [GB] | Number of Dask workers | Number of threads/CPUs | Dask scheduler type |
|---|---|---|---|---|---|---|
| v2.8.0 (threaded) on 1/16 node | v2.8.0 | 1/16 | 16 | 8 | 8 | thread-based |
| v2.8.0 (threaded) on 1 node | v2.8.0 | 1 | 256 | 128 | 128 | thread-based |
| v2.11.0 (threaded) on 1 node | v2.11.0 | 1 | 256 | 128 | 128 | thread-based |
| v2.11.0 (distributed) on 1/16 node | v2.11.0 | 1/16 | 16 | 2 | 8 | distributed |
| v2.11.0 (distributed) on 1 node | v2.11.0 | 1 | 256 | 32 | 128 | distributed |
| v2.11.0 (distributed) on 2 nodes | v2.11.0 | 2 | 512 | 64 | 256 | distributed |

of ESMValTool, and (2) different hardware setups. For (1), we compare the latest ESMValTool version 2.11.0 (with ESMVal-Core v2.11.1, Iris v3.11.0, and Dask v2024.11.2) against the reference version 2.8.0 (with ESMValCore v2.8.0, Iris v3.5.0, and Dask v2023.4.1), which is the last version of ESMValTool with no support of Dask distributed schedulers. Using older versions than v2.8.0 as a reference would yield even larger computational improvements. For (2), we compare running ESMValTool on different hardware setups on the state-of-the-art BullSequana XH2000 supercomputer *Levante* operated by the Deutsches Klimarechenzentrum (DKRZ) (see https://docs.dkrz.de/doc/levante/, last access: 10 December 2024). An overview of all investigated configurations is given in Table 1. Since all setups use the exact same CPU type (only the number of cores used varies) and memory per thread/CPU (2 GB), the different runtimes are directly comparable. When using local hardware like a standard laptop, the memory usage of the OS has to be accounted for. Thus, to reproduce our results here, a machine with a total memory of $16\,\text{GB} + x$ would be necessary (for example, $x \geq 4\,\text{GB}$ on a modern Ubuntu system). However, with a reduced number of workers and/or memory per worker, machines with 16 GB of memory in total could be sufficient (depending on the use case; not tested).

In order to present the results in a machine-agnostic way, we focus on the number of nodes of the specific setups instead of the exact amount of memory. This allows us to transfer the results to other machines than DKRZ's Levante and also to directly compare this to the speedup factor for an analysis of the computational efficiency, a number that measures the relative performance of two systems processing the same problem (see below).

To measure computational performance, we use the speedup factor $s$ as the main metric. The speedup factor of a new setup relative to a reference setup $REF$ can be derived from the corresponding execution times $t$ and $t_{REF}$:

$$s = \frac{t_{REF}}{t} \tag{1}$$





Values of $s > 1$ correspond to a speedup relative to the reference setup, values of $s < 1$ to a slowdown. For example, a speedup factor of $s = 2$ means that a run finishes twice as fast in the new setup compared to the reference setup.

A further metric we use is the scaling efficiency $e$, which is calculated as the resource usage (measured in node hours) in the reference setup divided by the resource usage in the new setup:

$$e = \frac{n_{REF} \cdot t_{REF}}{n \cdot t} = s \cdot \frac{n_{REF}}{n}, \qquad (2)$$

Here, $n$ and $n_{REF}$ are the number of nodes used in the new and reference setup, respectively. Values of $e > 1$ indicate that the new setup uses less computational resources than the reference setup, values of $e < 1$ that the new setup uses more

resources than the reference setup. For example, a scaling efficiency of $e = 2$ means that using the new setup only uses half of the computational resources (here, node hours) than the reference setup.

We deliberately ignore memory usage here since one aim of using Dask is to rather *optimize* memory usage instead of simply minimizing it, i.e., to use as much memory as possible (constrained by the system's availability and/or the user's configuration of Dask). A low memory usage is not necessarily desirable but could be the result of a non-optimal configuration. For example,

using a Dask configuration file designed for a standard laptop with 16 GB of memory (see e.g. Appendix A) would be very inefficient and result in higher runtimes on HPC system nodes with 256 GB of memory, since at most 6.25% of the available memory would be used. Nevertheless, if low memory usage is crucial (e.g., on systems with limited memory), Dask distributed can be configured to take into account such restrictions, which enables processing of data that are too large to fit into memory (out-of-core computing). Without this feature, processing large data sets on small machines would not be possible.

## 3.1    Multi-model analysis

The first example illustrates a typical use case of ESMValTool: the evaluation of a large ensemble of CMIP6 models. Here, we focus on the time series from 1850–2100 of the global mean sea surface temperature (SST) anomalies in the shared socioeconomic pathways SSP1-2.6 and SSP5-8.5 from a total of 238 ensemble members of CMIP6 models (O'Neill et al., 2016), as shown in Figure 9.3a of the IPCC's AR6 (Fox-Kemper et al., 2021). We reproduce this plot with ESMValTool

in Figure 2. As illustrated by the shaded area (corresponding to the 17%–83% model range), the models agree well over the historical period (1850–2014), but show larger differences for the projected future time period (2015–2100). A further source of uncertainty is the emission uncertainty (related to the unknown future development of human society), which is represented by the diverging blue and red lines, corresponding to the SSP1-2.6 and SSP5-8.5 scenarios, respectively. SSP1-2.6 is a sustainable low-emission scenario with a relatively small SST increase over the 21st century, while SSP5-8.5 is a fossil fuel–intensive

high-emission scenario with very high SSTs in 2100. For details, we refer to Fox-Kemper et al. (2021).

The input data of Figure 2 are 3-dimensional SST fields (time, latitude, longitude) from 238 ensemble members of 33 different CMIP6 models, adding up to around 230 GB of data in total. The following preprocessors are used in the corresponding ESMValTool recipe (in this order):

1. `anomalies`: calculate anomalies relative to the 1950—1980 climatology.

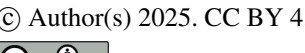



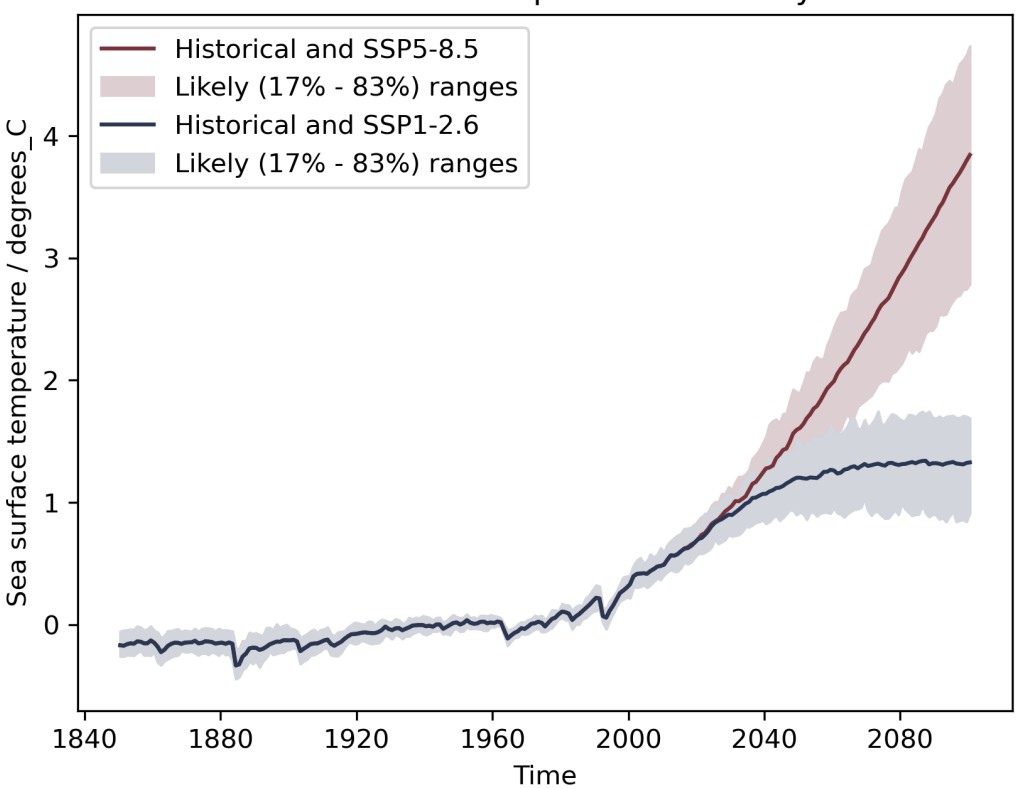

**Figure 2.** Time series of global and annual mean sea surface temperature anomalies in the shared socioeconomic pathways SSP1-2.6 and SSP5-8.5 relative to the 1950—1980 climatology, calculated from 238 ensemble members of CMIP6 models. The solid lines show the multi-model mean; the shading the likely (17%–83%) ranges. Similar to Figure 9.3a of the IPCC's AR6 (Fox-Kemper et al., 2021).

2. `area_statistics`: calculate global means.

    3. `annual_statistics`: calculate annual means.

    4. `ensemble_statistics`: calculate ensemble means for the models that provide multiple ensemble members.

    5. `multi_model_statistics`: calculate multi-model mean and percentiles (17% and 83%) from the ensemble means.

Table 2 shows the speedup factors and scaling efficiencies of ESMValTool runs to produce Figure 2 in different setups
(each entry has been averaged over two independent ESMValTool runs). We find that version 2.11.0 of ESMValTool on one node performs much better than the reference setup (version 2.8.0 on one node) with a speedup factor of 2.6 (i.e., reducing the runtime by more than 60%). By utilizing distributed computing with two nodes, the speedup factor for v2.11.0 further increases





**Table 2.** Speedup factors (see Equation 1) and scaling efficiencies (see Equation 2) of ESMValTool runs producing Figure 2 using different setups (averaged over two ESMValTool runs). A "-" denotes that the corresponding recipe run did not finish due to insufficient memory. Values in bold font correspond to the largest improvements. The speedup factors $s$ and scaling efficiencies $e$ are calculated relative to the setup "reference", which requires a runtime of approximately 3 hours and 27 minutes.

| Setup (see Table 1) | Speedup factor $s$ [1] | Scaling efficiency $e$ [1] |
| --- | --- | --- |
| v2.8.0 (threaded) on 1/16 node | - | - |
| v2.8.0 (threaded) on 1 node [reference] | 1.0 | 1.0 |
| v2.11.0 (distributed) on 1/16 node | 0.99 | **16** |
| v2.11.0 (distributed) on 1 node | 2.6 | 2.6 |
| v2.11.0 (distributed) on 2 nodes | **3.2** | 1.6 |

to 3.2. Using only 1/16 node, the recipe runs out of memory in ESMValTool v2.8.0 but successfully finishes in v2.11.0 with a speedup factor of 0.99. Even though the runtime slightly increased here relative to the reference version, the resource usage is much more optimal, which is illustrated by the scaling efficiency of 16, i.e., the recipe can be run almost 16 times as often in the new setup than in the reference setup for the same computational cost. For the other v2.11.0 setups using one and two nodes, the scaling efficiencies are much lower, though, with values of 2.6 and 1.6, respectively.

## 3.2 High-resolution data

A further common use case of ESMValTool is the analysis of high-resolution data. Here, we process 10 years of daily near-surface air temperature data (time, latitude, longitude) from the CMIP6 model NICAM16-9S (Kodama et al., 2021) with an approximate horizontal resolution of 0.14°x0.14°. We use results from the *highresSST-present* experiment (Haarsma et al., 2016), which is an atmosphere-only simulation of the recent past (1950–2014) with all natural and anthropogenic forcings, and SSTs and sea-ice concentrations prescribed from HadISST (Rayner et al., 2003). In total, this corresponds to 35 GB of input data. In our example recipe, which is illustrated in Figure 3, we calculate monthly climatologies averaged over the time period 1950–1960 of the near-surface air temperature over land grid cells for three different regions: Northern Hemisphere (NH) extratropics (30°N–90°N), Tropics (30°N–30°S), and Southern Hemisphere (SH) extratropics (30°S–90°S). The land-only near-surface air temperature shows a strong seasonal cycle in the extratropical regions (with the NH extratropical temperature peaking in July and the SH extratropical temperature peaking in January), but only very little seasonal variation in the Tropics. NH land temperatures are on average higher than SH land temperatures due to the different distribution of land masses in the hemispheres (for example, the South Pole is located on the large land mass of Antarctica [included in the calculation], whereas the North Pole is located over the ocean [excluded from the calculation]).

The ESMValTool recipe to produce Figure 3 uses the following preprocessors (in this order):

1. `mask_landsea`: mask out ocean grid cells so only land remains.



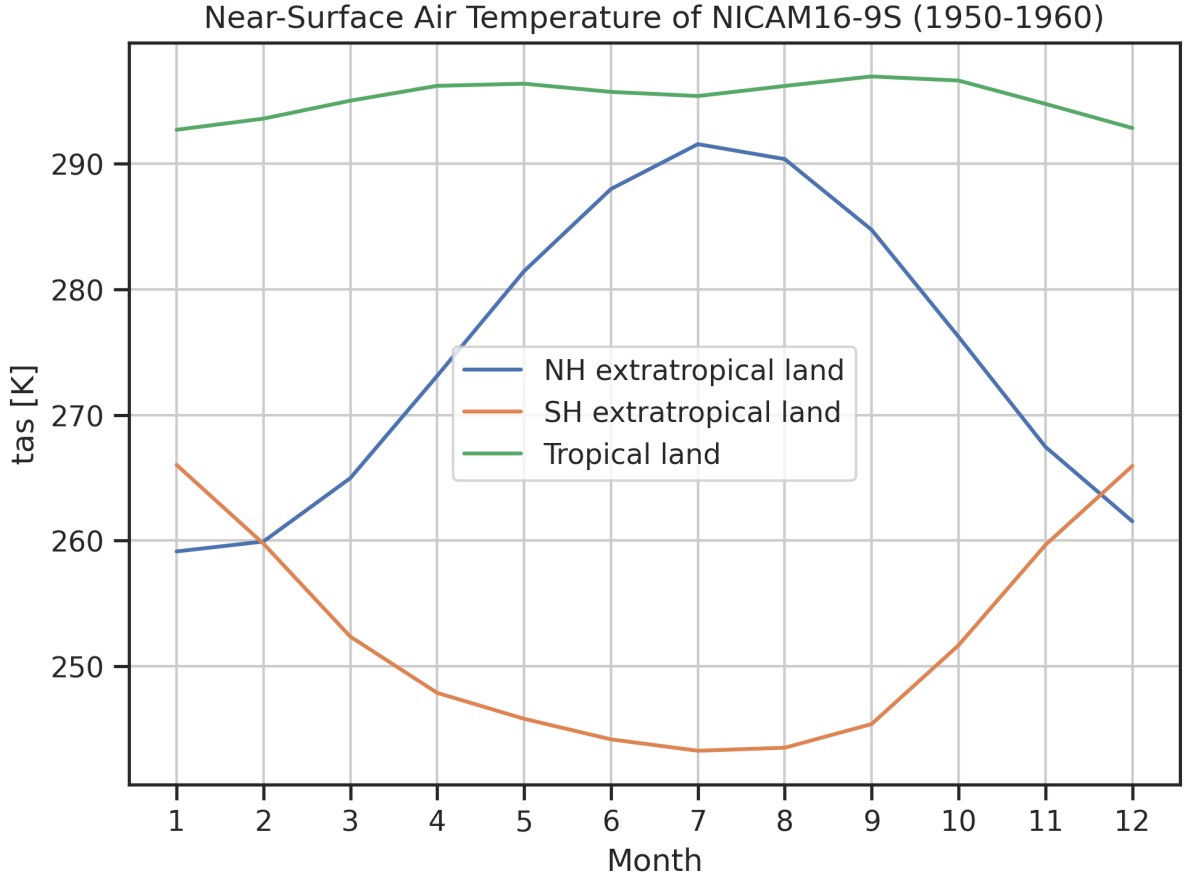

**Figure 3.** Monthly climatologies of near-surface air temperature (land grid cells only, averaged over the time period 1950–1960) for different geographical regions as simulated by the CMIP6 model NICAM16-9S using the *highresSST-present* experiment. The regions are defined as follows: Northern Hemisphere (NH) extratropics: 30°N–90°N, Tropics: 30°N–30°S, and Southern Hemisphere (SH) extratropics: 30°S–90°S.

   2. `extract_region`: cut out the desired regions (NH extratropics, Tropics, and SH extratropics)

3. `area_statistics`: calculate area mean over the corresponding regions.

   4. `monthly_statistics`: calculate monthly-means from daily input fields.

   5. `climate_statistics`: calculate the monthly climatology.

The corresponding speedup factors and scaling efficiencies for running this ESMValTool recipe with different setups are listed in Table 3. Similar to Table 2, the values are averaged over two ESMValTool runs of the same recipe. Again, ESMValTool

version 2.11.0 performs much better than its predecessor version 2.8.0. We find a massive speedup factor and scaling efficiency





**Table 3.** Speedup factors (see Equation 1) and scaling efficiencies (see Equation 2) of ESMValTool runs producing Figure 3 using different setups (averaged over two ESMValTool runs). A "-" denotes that the corresponding recipe run did not finish due to insufficient memory. Values in bold font correspond to the largest improvements. The speedup factors $s$ and scaling efficiencies $e$ are calculated relative to the setup "reference", which requires a runtime of approximately 40 minutes.

| Setup (see Table 1) | Speedup factor $s$ [1] | Scaling efficiency $e$ [1] |
|---|---|---|
| v2.8.0 (threaded) on 1/16 node | - | - |
| v2.8.0 (threaded) on 1 node [reference] | 1.0 | 1.0 |
| v2.11.0 (distributed) on 1/16 node | 2.5 | **40** |
| v2.11.0 (distributed) on 1 node | 25 | 25 |
| v2.11.0 (distributed) on 2 nodes | **36** | 18 |

of 25 when using ESMValTool v2.11.0 on one node compared to the reference setup (v2.8.0 on one node). When using two entire nodes, the speedup factor further increases to 36 with a drop in the scaling efficiency to 18. This demonstrates the powerful distributed computing capabilities of ESMValTool v2.11.0. Even when only 1/16 node is available, we find a moderate speedup factor of 2.5, but a very high scaling efficiency of 40, indicating that this setup uses almost 40 times less

computational resources than the reference version. Similar to the multi-model analysis presented in the previous section, the recipe fails to run with v2.8.0 on 1/16 node due to insufficient memory (total memory: 16 GB; total input data size: 35 GB).

### 3.3 Commonly used preprocessing operations

In this section, the performance of individual preprocessing operations in an idealized setup is shown. For this, the following five preprocessors are applied to monthly-mean, vertically resolved air temperature data (time, pressure level, latitude, longi-

tude) of 20 years of historical simulations (1995–2014) from 10 different CMIP6 models: (1) `area_statistics` (calculates global area means), (2) `climate_statistics` (calculates climatologies), (3) `regrid` (regrid the input data to a regular 5°x5° grid), (4) `amplitude` (calculates seasonal cycle amplitudes), and (5) `extract_levels` (interpolate the data to the pressure levels 950 hPa, 850 hPa, 750 hPa, 500 hPa, and 300 hPa). These five preprocessors have been chosen since they are among the most commonly used preprocessors across all available recipes (currently, more than 60 different preprocessors are

available in ESMValTool). Figure 4 shows the speedup factors and scaling efficiencies of these preprocessors. In contrast to the previous two sections, this considers the times to run these preprocessors, not the entire runtimes of the corresponding ESMValTool recipes, which can be considerably larger due to computational overhead like searching for input data, data checks, etc.

  As shown in Figure 4a, ESMValTool version 2.11.0 with a distributed scheduler consistently outperforms its predeces-

sor version 2.8.0 with the thread-based scheduler. The speedup factors range, depending on the preprocessor and number of nodes, from 1.2 for `amplitude` on 1/16 node to 22 for `extract_levels` on one node. In v2.8.0, using more computa-



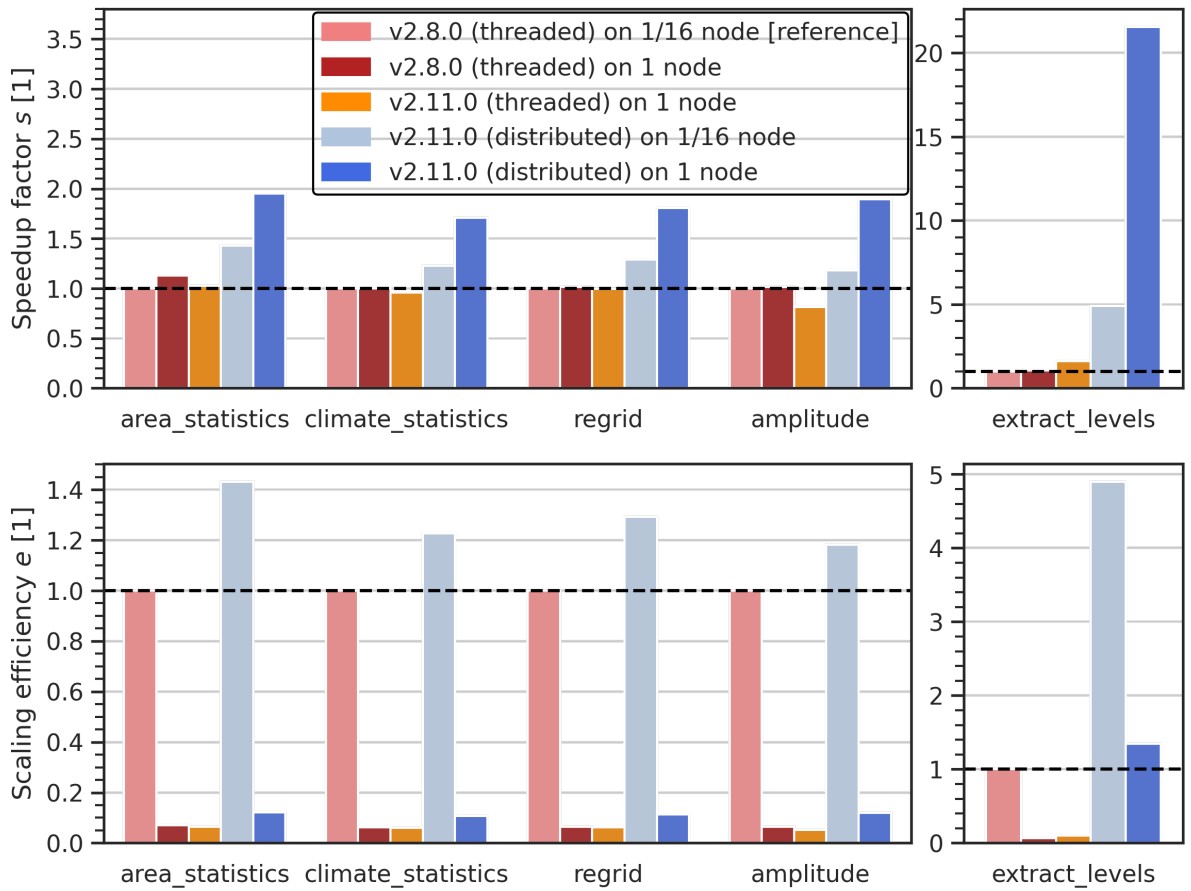

**Figure 4.** (a) Speedup factors (see Equation 1) and (b) scaling efficiencies (see Equation 2) of individual preprocessor runs in different setups (see legend). The speedup factors $s$ and scaling efficiencies $e$ are calculated relative to the setup "reference". The input data are monthly-mean vertically resolved air temperature (time, pressure level, latitude, longitude) for 20 years of historical simulations (1995–2014) from 10 different CMIP6 models. The black dashed lines indicate no improvement relative to the reference setup and values above/below these lines correspond to better/worse performance.

tional resources only slightly reduces the runtime, while in v2.11.0 a substantial speedup can be achieved, especially for the `extract_levels` preprocessor (from $s = 4.9$ on 1/16 node to $s = 22$ on one node, which itself is a speedup factor of 4.5). For all analyzed preprocessors except `extract_levels`, using the default thread-based scheduler in v2.11.0 gives speedup factors of $\approx 1$, which indicates similar runtimes as in v2.8.0. For `extract_levels`, we find a speedup factor of 1.6 in this setup.

In terms of computational resources, it is much more expensive to run the analyzed preprocessors on one node instead of 1/16 node, which is illustrated in Figure 4b. Only the setup v2.11.0 (distributed) on 1/16 node consistently shows scaling




efficiencies > 1 across all preprocessors. For `extract_levels`, also the setup v2.11.0 (distributed) on one node gives a
scaling efficiency > 1 (1.3), whereas all other analyzed preprocessors show scaling efficiencies ≪ 1 in this setup.

## 4   Discussion

The results above show that ESMValTool's performance is considerably improved in v2.11.0 compared to v2.8.0 through the
consistent use of Dask within ESMValTool's core packages, i.e., ESMValCore and Iris.

Only small improvements were found for preprocessing operations that were already using Dask in ESMValTool v2.8.0.
These include the `area_statistics` ($s = 1.4$ on 1/16 node; $s = 1.9$ on one node), `climate_statistics` ($s = 1.2$ on
1/16 node; $s = 1.7$ on one node), `regrid` ($s = 1.3$ on 1/16 node; $s = 1.8$ on one node), and `amplitude` ($s = 1.2$ on 1/16
node; $s = 1.9$ on one node) preprocessors (see Figure 4). The speedup factors in parentheses correspond to v2.11.0 (distributed
scheduler) vs. v2.8.0 (thread-based scheduler) in the given setup. Here, the performance improvements can be traced back
to the use of a Dask distributed scheduler instead of the default thread-based scheduler, since using the latter gives similar
runtimes in v2.11.0 and v2.8.0 (see orange bars in Figure 4). Especially with many workers, the distributed scheduler can
execute the task graph much more efficiently, which leads to higher speedup factors. The corresponding scaling efficiencies
are ≪ 1 for all setups using one node, indicating that running these preprocessors on one node is not desirable if the aim is to
minimize the computational resources used. The reasons for these low scaling efficiencies are the low speedup factors, which
do not compensate the increase in resource consumption when moving from 1/16 node to one full node.

Large improvements can be found for preprocessors which were not yet fully taking advantage of Dask in v2.8.0, but
mostly relied on regular Numpy arrays instead. In addition to the speedup gained by using a Dask distributed scheduler, the
calculations can now be executed in parallel (and, if desired, also distributed across multiple nodes or machines), which results
in an additional massive performance improvement, in particular in a large memory setup (e.g., 256 GB per node) with a high
number of workers. An important example here is the preprocessor `extract_levels` ($s = 4.9$ on 1/16 node; $s = 22$ on one
node) for which we find scaling efficiencies $e > 1$ in all setups that use a distributed scheduler. This means that the increased
resource consumption is compensated with much shorter run times. This performance improvement leads to considerably more
efficient and faster processing of vertically resolved (and thus much more computationally expensive) variables.

Other preprocessor functions that benefit heavily from the consistent usage of Dask are `multi_model_statistics`
(see Section 3.1) and `mask_landsea` (see Section 3.2). As shown in Tables 2 and 3, the corresponding recipes fail to run
on 1/16 node in v2.8.0 because the aforementioned preprocessors produce out-of-memory errors. The reason for this is that
ESMValTool v2.8.0 tries to load data sets that are larger than the available memory. Only the optimized implementation of
these preprocessors with Dask allows running the corresponding recipes in the small-memory setup via out-of-core computing.
Especially for the example of evaluating a high-resolution model (see Section 3.2), this is very effective: ESMValTool v2.11.0
on 1/16 node is 2.5 times faster and 40 times more efficient (in terms of computational resources) than ESMValTool v2.8.0 on
one full node. When using an entire node with v2.11.0, this speedup further increases to $s = 25$. With distributed computing
using two nodes, the speedup factor rises to $s = 36$. Since this increase in the speedup factor is smaller than the theoretical limit





possible by the higher resource usage, the scaling efficiencies drop to $e = 25$ and $e = 18$ for one and two nodes, respectively. The main reason for these massive performance gains is that most of the runtime of this recipe is spent on the actual preprocessing calculations (the analysis contains only one large data set), which can be highly optimized with Dask.

285 For the multi-model analysis example from Section 3.1, the speedup factors and scaling efficiencies are smaller than in case of the high-resolution data set ($s = 0.99$ and $e = 16$ on 1/16 node; $s = 2.6$ and $e = 2.6$ on one node; $s = 3.2$ and $e = 1.6$ on two nodes), but still substantial. In contrast to the high-resolution model evaluation example, this recipe contains a large number of data sets (here, 238), resulting in a large contribution of serial operations, such as loading and processing coordinate data (approximately 50% of runtime) and finding input files (typically only a few percent of the runtime). Loading coordinate

data can be relatively time-consuming because it requires reading small amounts of data from many files, which the Lustre file system used by DKRZ's HPC system Levante is not optimized for. Moreover, climate model data are usually available in netCDF/HDF5 format with data written by the climate models typically at each output time step. This can result in netCDF chunk sizes that are far from optimal for reading. For example, reading a compressed time coordinate containing 60,000 points with a netCDF chunk size of 1 (total uncompressed size 0.5 MB) as used for $\sim 20$ years of 3-hourly model output, can take

up to 15 seconds on Levante. Finding files, while typically relatively fast, can become time-consuming when there is a need to query the servers of the Earth System Grid Foundation (ESGF) to find and download input data or when there is a high load on the shared Lustre file system. All these serial operations cannot be sped up by using more nodes/Dask workers. Thus, an increase in the number of nodes/Dask workers by a factor of $n$ will always lead to a speedup factor of $n - x$ with $x \in (0, n)$, where the size of $x$ depends on the type of analysis conducted in the recipe. For example, the multi-model recipe has a much

higher $x$ than the high-resolution recipe. Consequently, by Equation 2, the scaling efficiency will always be smaller in larger setups, which also becomes apparent in Tables 2 and 3, and Figure 4.

## 5 Summary and Outlook

This paper describes the large improvements in ESMValTool's computational efficiency achieved through continuous optimization of the code over the last years that are now available in the release of version 2.11.0. The consistent use of the Python

package Dask within ESMValTool's core libraries, ESMValCore and Iris, improves parallel computing (parallelize computations) and out-of-core computing (process data that are too large to fit entirely into memory), and enables distributed computing (distribute computations across multiple nodes or interconnected machines). With these optimizations, we find substantially shorter runtimes for real-world ESMValTool recipes executed on a single HPC node with 256 GB of memory, ranging from 2.6 times faster runs in a multi-model setting up to 25 faster runs for the processing of a single high-resolution model. Using

two nodes with v2.11.0 (512 GB of memory in total), the speedup factors further improve to 3.2 and 36, respectively. These enhancements are enabled by the new optimized parallel and distributed computing capabilities of ESMValTool v2.11.0, and could be improved even further by using more Dask workers. While both example recipes fail to run in the small setup (1/16 node; 16 GB of memory) with ESMValTool version v2.8.0, they can be run successfully with v2.11.0 by using the new out-of-core computing capabilities of the tool. This enables running ESMValTool recipes that process very large data sets even





on smaller hardware like a standard laptop. The more detailed analysis of individual frequently used preprocessor functions shows similar improvements. We find speedup factors of 1.2 to 22 for different preprocessors, depending on the degree of optimization of the preprocessor in the old ESMValTool version 2.8.0.

In addition to these massive speedups, evaluation runs with ESMValTool v2.11.0 can be configured to use less computational resources by using e.g. only part of an HPC node. Therefore, if the target is optimizing the resource usage instead of the runtime,

it is advisable to use a small setup. As a positive side effect, this also minimizes the power consumption and thus the carbon footprint of ESMValTool runs. For example, using 1/16 node, the two real-world recipes use 16 (multi-model analysis) and 40 (high-resolution model analysis) times less resources compared to ESMValTool v2.8.0 running on one full node. As mentioned before, running v2.8.0 on 1/16 node is not possible due to out-of-memory errors. It should be noted here that ESMValTool v2.8.0 usually needs somewhat less memory than available on an entire node, so the scaling efficiencies would be slightly

lower if this minimal-memory setup was considered as reference. Such minimal-memory setups, however, would have to be created for each individual recipe and are thus not feasible in practice. This is why typically a full HPC node was used with ESMValTool v2.8.0.

In ESMValTool v2.11.0, around 90% of the available preprocessor functions are consequently using Dask and can be run very efficiently with a Dask distributed scheduler. Thus, current and future development efforts focus on the optimization of the

remaining not yet optimized 10% of preprocessors within ESMValCore and/or Iris. A particular preprocessor that is currently being optimized is regridding. Regridding is an essential processing step for many diagnostics, especially for high-resolution data sets and/or model data on irregular grids (e.g., Schlund et al., 2023). Currently, support for various grid types and different algorithms is improved, in particular within the Iris-esmf-regrid package (Worsley, 2024), which provides efficient regridding algorithms that can be directly used with ESMValTool recipes. Further improvements could include running all metadata and

data computations on Dask workers, which would significantly speed up recipes that use many different data sets and thus require significant metadata handling (like the multi-model example presented in Section 3.1). ESMValTool is also expected to benefit strongly from further optimizations of the Iris package. For example, first tests with an improved data concatenation recently introduced in Iris' development branch show very promising results.

All developments presented in this study will strongly facilitate evaluation of complex, high-resolution and large ensem-

bles of ESMs, in particular of the upcoming generation of climate models from CMIP7 (for example, ESMValTool will be part of the Rapid Evaluation Workflow [REF]; see https://wcrp-cmip.org/cmip7/rapid-evaluation-framework/, last access: 10 December 2024). With this, ESMValTool is getting ready to be able to provide analyses that can deliver valuable input for the Seventh Assessment Report of the IPCC (planned to be fully completed by late 2029; see https://ipcc.ch/2024/01/19/ ipcc-60-ar7-work-programme/, last access: 10 December 2024), which will ultimately provide valuable insights into climate

change and help developing and assessing effective mitigation and adaptation strategies.

*Code and data availability.* Supplementary material for reproducing the analyses of this paper (including ESMValTool recipes) is publicly available on Zenodo at https://doi.org/10.5281/zenodo.14361733. The improvements described here are available since ESMValTool v2.11.0.



ESMValTool v2 is released under the Apache License, VERSION 2.0. The latest release of ESMValTool v2 is publicly available on Zenodo at https://doi.org/10.5281/zenodo.3401363 (Andela et al., 2024a). The ESMValCore package, which is installed as a dependency of ESM-
ValTool v2, is also publicly available on Zenodo at https://doi.org/10.5281/zenodo.3387139 (Andela et al., 2024b). ESMValTool and ESM-ValCore are developed on the GitHub repositories available at https://github.com/ESMValGroup (last access: 10 December 2024). The Iris package, which is a dependency of ESMValTool and ESMValCore, is publicly available on Zenodo at https://doi.org/10.5281/zenodo.595182 (Iris contributors, 2024). Detailed user instructions to configure Dask within ESMValTool can be found in the ESMValTool documentation at https://docs.esmvaltool.org/projects/ESMValCore/en/v2.11.1/quickstart/configure.html#dask-configuration (last access: 10 December 2024)
and in the ESMValTool tutorial at https://tutorial.esmvaltool.org/ (last access: 10 December 2024). The documentation is recommended as a starting point for new users and provides links to further resources. For further details, we refer to the general ESMValTool documentation available at https://docs.esmvaltool.org/ (last access: 10 December 2024) and the ESMValTool website (https://esmvaltool.org/, last access: 10 December 2024).

CMIP6 model output required to reproduce the analyses of this paper is available through the Earth System Grid Foundation (ESGF;
https://esgf-metagrid.cloud.dkrz.de/search/cmip6-dkrz/, last access: 10 December 2024). ESMValTool can automatically download these data if requested (see https://docs.esmvaltool.org/projects/ESMValCore/en/v2.11.1/quickstart/configure.html#esgf-configuration, last access: 10 December 2024.

## Appendix A: Example Dask configuration file for single machines

To run ESMValTool with a Dask distributed scheduler on a single machine, the following Dask configuration file could be
used:

```
# File ~/.esmvaltool/dask.yml
---
cluster:
  type: distributed.LocalCluster
  n_workers: 4
  threads_per_worker: 2
  memory_limit: 3 GB
```

This will spawn a Dask `distributed.LocalCluster` with 4 workers in total. Each worker uses 2 threads and has access to 3 GB of memory. These settings are well suited to run ESMValTool on a standard laptop with 16 GB of memory (4
GB are withheld for the operating system).

## Appendix B: Example Dask configuration file for HPC systems

To run ESMValTool with a Dask distributed scheduler on an HPC system that uses the Slurm workload manager (https://slurm.schedmd.com/, last access: 10 December 2024), the following Dask configuration file could be used:



```
     # File ~/.esmvaltool/dask.yml
380  ---
     cluster:
       type: dask_jobqueue.SLURMCluster
       queue: compute
       account: SLURM_account_name
cores: 128
       memory: 256GB
       processes: 64
       interface: ib0
       local_directory: /path/to/temporary/directory
n_workers: 64
       walltime: 08:00:00
```

Here, the `dask_jobqueue.SLURMCluster` will allocate 128 cores with 256 GB of total memory per job on *compute* nodes for 8 hours. The node allocation will be handled by Dask; no manual call to `sbatch`, `srun`, etc. is necessary. Each job should be cut up in 64 processes, and 64 workers in total are requested. Thus, this will launch exactly one job (`processes` is

the number of workers per job), which corresponds to the allocation of exactly one "compute" node on DKRZ's Levante. The other options here are a Slurm account for which the user can request resources, the network interface (here, the fast InfiniBand standard is used), and a local directory where temporary files can be stored.

*Author contributions.* MS designed the concept of this study, conducted the analysis presented in the paper, led the writing of the paper, and contributed to the ESMValTool, ESMValCore, and Iris source code. BA contributed to the concept of this study, and the ESMValTool,
ESMValCore, and Iris source code. JB contributed to the ESMValCore source code. RC contributed to the ESMValCore and Iris source code. BH contributed to the concept of this study, and the ESMValTool and ESMValCore source code. EH contributed to the ESMValTool, ESMValCore, and Iris source code. PK contributed to the ESMValTool, ESMValCore, and Iris source code. AL contributed to the concept of this study, and the ESMValTool and ESMValCore source code. BL contributed to the ESMValTool, ESMValCore, and Iris source code. SLT contributed to the ESMValTool, ESMValCore, and Iris source code. FN contributed to the ESMValCore and Iris source code. PP
contributed to the ESMValCore and Iris source code. VP contributed to the ESMValTool, ESMValCore, and Iris source code. SS contributed to the ESMValTool, ESMValCore, and Iris source code. SW contributed to the ESMValCore and Iris source code. MY contributed to the ESMValTool, ESMValCore, and Iris source code. KZ contributed to the ESMValTool, ESMValCore, and Iris source code. All authors contributed to the text.

*Competing interests.* Some authors are members of the editorial board of Geoscientific Model Development. The authors have no other
competing interests to declare.



*Acknowledgements.* The development of ESMValTool is supported by several projects. Funding for this study was provided by the European Research Council (ERC) Synergy Grant "Understanding and Modelling the Earth System with Machine Learning (USMILE)" under the Horizon 2020 research and innovation programme (Grant Agreement No. 855187). This project has received funding from the European Union's Horizon 2020 research and innovation programme under Grant Agreement No. 101003536 (ESM2025 — Earth System Models for the Future). This project has received funding from the European Union's Horizon Europe research and innovation programme under Grant Agreement No. 101137682 (AI4PEX — Artificial Intelligence and Machine Learning for Enhanced Representation of Processes and Extremes in Earth System Models). This project has received funding from the European Union's Horizon Europe research and innovation programme under Grant Agreement No. 824084 (IS-ENES3 — Infrastructure for the European Network for Earth System Modelling). This project has received funding from the European Union's Horizon Europe research and innovation programme under Grant Agreement No. 776613 (EUCP — European Climate Prediction system). The performance optimizations presented in this paper have been made possible by the ESiWACE3 (3rd phase of the Centre of Excellence in Simulation of Weather and Climate in Europe) Service 1 project. ESiWACE3 has received funding from the European High Performance Computing Joint Undertaking (EuroHPC JU) and the European Union (EU) under grant agreement No. 101093054. Views and opinions expressed are however those of the author(s) only and do not necessarily reflect those of the European Union, the European Climate, Infrastructure and Environment Executive Agency (CINEA). Neither granting authority can be held responsible for them. This research was supported by the BMBF under CAP7 project, Grant Agreement No. 01LP2401C. Support for Dask Distributed was added to ESMValCore as part of the ESMValTool Knowledge Development project funded by the Netherlands eScience Center in 2022/2023. We thank the natESM project for the support provided through the sprint service, which contributed to the ESMValTool developments and optimizations presented in this study. natESM is funded through the Federal Ministry of Education and Research (BMBF) under grant agreement No. 01LK2107A. Development and maintenance of Iris is primarily by the UK Met Office - funded by the Department for Science, Innovation and Technology (DSIT) - and by significant open-source contributions (see the other funding sources listed here). We acknowledge the World Climate Research Programme (WCRP), which, through its Working Group on Coupled Modeling, coordinated and promoted CMIP6. We thank the climate modeling groups for producing and making available their model output, the Earth System Grid Federation (ESGF) for archiving the data and providing access, and the multiple funding agencies who support CMIP and ESGF. This work used resources of the Deutsches Klimarechenzentrum (DKRZ) granted by its Scientific Steering Committee (WLA) under project IDs bd0854, bd1179 and id0853. We would like to thank Franziska Winterstein (DLR) for helpful comments on the manuscript.



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
