# Peer review of "Advanced climate model evaluation with ESMValTool v2.11.0 using parallel, out-of-core, and distributed computing"

_Geoscientific Model Development, 2024_

## Author Response (AR1)

**Answer to Anonymous Referee #1 (RC1)**

Reviewer comments are given in **bold**, our answers in blue.

**Thank you for this article. This was generally clearly written and it is a good reference for ESMValTool users and developers. I do appreciate that real-world and clearly defined use cases have been analysed. The benefits of the work to the global model evaluation community are also clearly articulated.**

**I am therefore happy to recommend this article for publication in GMD but please find some comments that could improve the quality of the article.**

We would like to thank the reviewer for taking the time to evaluate our work and for their helpful and constructive comments. We have revised our manuscript in light of these and the other reviewers' comments we have received. A pointwise reply is given below. Line numbers in the answers correspond to line numbers in the revised manuscript.

**One section in particular would benefit from some clarifications. I do find the section between lines 285 and 301 explaining scalability in Table 2 quite difficult to follow.**

**The explanation about coordinate files is difficult to understand. On one hand, it is referred to as a serial operation (line 288), on the other hand it is said to be based on reading many small files (line 290)– which could lead to easy parallelism. Line 289 it is written that loading and processing coordinates takes 50% of runtime but without stating exactly on which case. Is it the reference case? It cannot be 50% equally on all experiments. Could this be explained more clearly ?**

We added an explanation of why parallelizing these operations is not trivial:

L.268–276: "*On the other hand, the multi-model analysis involves a large number of data sets and files (here, data from 238 ensemble members of 33 different CMIP6 models scattered over 3708 files), which requires a high number of serial metadata computations like loading the list of available variables in files and processing coordinate arrays (e.g., time, latitude, longitude), resulting in a much larger proportion of serial operations. Consequently, the maximum possible speedup factor and thus the actual speedup factor is a lot smaller in this example. Parallelizing these metadata operations is more challenging because it requires performing operations on Iris cubes (i.e., the fundamental Iris objects representing multi-dimensional data with metadata) on the Dask workers instead of just replacing Numpy arrays with Dask arrays. First attempts at this were not successful due to issues with the ability of underlying libraries (Iris and ESMPy) to run on Dask workers, but we aim to address those problems in future releases.*"

In addition, we do not mention anymore how much of the runtime is spent on loading and processing coordinates because this only based on a rough estimate and might lead to confusion.

**Line 296 says that some parts of the application profile are sensitive to Lustre load and ESGF servers connectivity. This begs the question: can we trust the timings presented in Table 2 and if yes - why?**

The results presented in Section 3 of our paper do not depend on connectivity to ESGF servers since all data is available locally; thus, there is no need to query ESGF servers or to download any data. We tried to minimize the influence of the Lustre file system on our results by repeating the analyses several times and/or running them during periods with little user activity (e.g., over night or on weekends). Since the fluctuations of the run times when using the same setup are much smaller than the differences between the run times of different setups, we are confident that our results are robust.

To emphasize that these two issues do not affect our results presented here, we removed the corresponding sentence in the revised manuscript and added the following clarifications to Sections 3.1 and 3.2, respectively:

L.192–194: " *Each entry here has been averaged over two independent ESMValTool runs to minimize the effects of random runtime fluctuations. Since the runtime differences within a setup are much smaller than the differences between different setups, we are confident that our results are robust.* "

L.221–223: " *Similar to Table 2, the values are averaged over two ESMValTool runs of the same recipe. The runtime differences within setups are again much smaller than the differences between different setups, indicating that the results are robust.* "

**Lines 298-302, this is a canonical example of Amdhal's law, but it is not completely correctly stated in the text (for example scalability >1 should remain true analytically).**

To properly reflect Amdahl's law, we rephrased the beginning and end of the corresponding paragraph:

L.263–268: " *The high-resolution analysis achieves significant performance gains because it uses data from only a single ensemble member of one climate model loaded from just 11 files. Consequently, most of the runtime of this workflow is spent on preprocessing calculations on the array representing near-surface air temperature, a task that can be highly optimized using Dask. In other words, the proportion of operations that can be parallelized is high, leading to a large maximum possible speedup factor (Amdahl's law). For example, if 90% of the code can be parallelized, the maximum possible speedup factor is 10 since 1/10 of the code cannot be sped* *

*up.* "

L.282–286: " *Amdahl's law also provides a theoretical explanation why the scaling efficiencies decrease when the number of HPC nodes is increased (see Tables 2 and 3). Due to the serial part of the code that cannot be parallelized, the speedup factor s will always grow slower than the number of HPC nodes n, resulting in a decrease of the scaling efficiencies e with rising n (see Equation 2). In the limit of infinite nodes n → ∞, s approaches a finite value (the aforementioned maximum possible speedup factor); thus, e → 0.* "

**Answer to Anonymous Referee #2 (RC2)**

Reviewer comments are given in **bold**, our answers in blue.

**Thank you for your submission. ESMValTool is clearly part of the production workflow for evaluating the output of Earth system models, and therefore its optimization is of significant importance to the climate community. The paper is well written and establishes a clear performance improvement of version 2.11.0 over the previous 2.8.0 version, The significant improvements are the possibility for out-of-core computing, which allows 2.11.0 to run on configurations where 2.8.0 reports out-of-memory errors, and the support for distributed execution. The paper clearly illustrates the resulting performance increase.**

We would like to thank the reviewer for taking the time to evaluate our work and for their helpful and constructive comments. We have revised our manuscript in light of these and the other reviewers' comments we have received. A pointwise reply is given below. Line numbers in the answers correspond to line numbers in the revised manuscript.

**Having said that, there are some key deficiencies in the methodology. In particular, the scaling efficiency metric provided in Equation (2) leads to confusion in the subsequent performance analysis, particularly in runs using "1/16 node". For example, the remarkable scaling efficiencies in Table 2 (e.g., 16) and Table 3 (e.g., 40) are in the cases where v.2.11.0 can run on 1/16-node where v2.8.0 on 1/16-node runs into memory limitations therefore cannot be used as the reference. These high values are an artifact of comparing the run time of v2.8.0 on a fully occupied node, but assuming that v2.11.0 is taking only one sixteenth of the node while all the other cores are doing effective work. However, most schedulers, particularly on DKRZ Levante, will not allow the exploitation of cores at this fine granularity, and thus the high efficiencies for the 1/16-node case are not realizable. The confusion increases in Figure 4 where v2.8.0 can be run on 1/16-node and therefore becomes the reference. The dramatically \*low\* efficiencies on a full node at the bottom of Figure 4 are again an artifact of the assumption that the other fifteen cores of the node could actually be doing effective work, which they realistically cannot.**

**My recommendation would be either to explain how all cores on the node can be sensibly occupied when ESMValTool occupies only 1/16-node, or to avoid the 1/16-node results entirely and concentrate of the objective improvements in the 1 and 2 node cases. The findings for the 1-node case are (1) v2.11.0 (threaded) is minimally slower or faster than v2.8.0 (threaded), but v2.11.0 (distributed) is signicantly faster, e.g., 1.7 - 2.0x in many cases, and 22x in the exceptional case of "extract_levels". The scaling efficiency should only be used in the case of comparing single node execution to multiple node (in this case 2), which essentially then becomes a strong**

**scaling analysis. There are still remarkable improvements to report, in particular the case the authors recount in lines 280-284.**

We agree that by using different reference setups for the different sections (1 node for Sections 3.1 and 3.2, 1/16 node for Section 3.3), the corresponding results can be confusing and difficult to interpret. Thus, as recommended by the reviewer, we removed the 1/16 node setup altogether and only consider 1- and 2-node setups for the analysis of the speedup factors and scaling efficiencies. The reference setup is now "v2.8.0 on one HPC node" for all sections. Instead of the 1/16 node setup, we run ESMValTool on a personal laptop with 16 GB of RAM to demonstrate ESMValTool's out-of-core computation capabilities. This has two main advantages: (1) the consistent reference setup allows a much easier comparison of the results of the different sections, and (2) using an actual laptop demonstrates that ESMValTool can be properly used without access to an HPC system. Since this change affects a large number of lines in the manuscript, we do not provide a detailed list of changed lines here but would like to refer to the "track changes" version of the manuscript instead.

We would like to point out that at least in theory when using a shared node on Levante, the resources of a single node can actually be properly shared between different jobs and/or users. For example, the following command can be used to request interactive access to 16 CPUs of a shared node:

```
$ salloc --partition=shared --cpus-per-task=16 --mem-per-cpu=1940M
```

On this node, asking for the number of processes available to the current thread with `nproc` gives 16, and analyzing the resource usage afterwards reveals that only 1/16 of the node hours compared to a full node have been consumed (=16/256 of the available CPUs). Furthermore, checking the running processes on the node with `htop` suggests that the node is indeed shared with other users.

In the manuscript, we now mention that computational costs can be further reduced by using a shared HPC node:

L.328–330: " *To further reduce computational costs on an HPC system, ESMValTool can be configured to run only on a shared HPC node using only parts of the node's resources. This reduces the influence of code that cannot be parallelized and thus optimizes the scaling efficiency (Amdahl's law; see Section 4).* "

**In spite of my concerns about the methodology, this paper does illustrate solid optimizations of a production tool which is central to data analysis. As you point out in lines 55-58, it is a key responsibility of our community to minimize the**

**impact and carbon footprint of climate computing. But you must also mention and put into perspective that the vast majority of the footprint is coming from the simulations themselves rather than diagnostic tools such as ESMValTool.**

Good point. We changed this to:

L.55–58: " *Second, minimizing the usage of computational resources reduces energy demand and the carbon footprint of HPC and data centers, which are expected to have a steadily increasing contribution to the total global energy demand in the upcoming years (Jones, 2018). Having said this, it should be noted that producing the actual ESM simulations requires much more computational resources than their evaluation with tools like ESMValTool.* "

**Answer to Anonymous Referee #3 (RC3)**

Reviewer comments are given in **bold**, our answers in blue.

**The manuscript describes the new advancement in the ESMValTool, which is one of widely used open-source Earth System model (ESM) evaluation tools. The new advancement focuses on the parallel optimization of the code to enable speed up of the code in parallel computation environment. The manuscript demonstrates that there was considerable speed up in the new version via active employment of Dask (Python library for parallelization). Considering the massive data size of ESMs, for example CMIP, it is definitely beneficial to have efficient computation capability especially when the code has to run for multiple datasets (i.e., multiple models) to enable benchmarking and intercomparison. Speaking of that, I believe it is worth documenting such improvement of the tool. The manuscript is well organized and prepared, and the speed up was clearly shown. With that, I support the publication of the manuscript.**

We would like to thank the reviewer for taking the time to evaluate our work and their positive feedback. In light of the other reviewers' comments, we have revised our manuscript.